# Realistic Evaluation of Semi-Supervised Learning Algorithms

**Avital Oliver**[*,†] **& Augustus Odena**[*] **& Colin Raffel**[*] **& Ekin D. Cubuk & Ian J. Goodfellow**
Google Brain
{avitalo,augustusodena,craffel,cubuk,goodfellow}@google.com

## Abstract

Semi-supervised learning (SSL) provides a powerful framework for leveraging unlabeled data when labels are limited or expensive to obtain. Approaches based on deep neural networks have recently proven successful on standard benchmark tasks. However, we argue that these benchmarks fail to address many issues that these algorithms would face in real-world applications. After creating a unified reimplementation of various widely-used SSL techniques, we test them in a suite of experiments designed to address these issues. We find that simple baselines which do not use unlabeled data can be competitive with the state-of-the-art, that SSL methods differ in sensitivity to the amount of labeled and unlabeled data, and that performance can degrade substantially when the unlabeled dataset contains out-of-class examples.

## This workshop paper is outdated.

## Please visit the updated, full version on arXiv: `https://arxiv.org/abs/1804.09170`

## 1 Introduction

The successes of deep neural networks on large-scale supervised learning problems come at a distinct cost; namely, creating these large datasets typically requires a great deal of human effort (to manually label examples), pain or risk (for medical datasets involving invasive tests) or financial expense (to hire labelers or build the infrastructure needed for domain-specific data collection). For many practical problems and applications, there are simply insufficient resources to create a sufficiently large labeled dataset, which limits the wide-spread adoption of deep learning techniques.

An attractive approach towards mitigating this issue is the framework of **semi-supervised learning** (SSL). SSL algorithms improve on the performance of supervised learning algorithms by also using unlabeled examples. Some recent results have shown that in certain cases, SSL approaches the performance of purely supervised learning, even when a substantial portion of the labels in a given dataset have been discarded.

These recent successes raise a natural question: Are SSL approaches applicable in "real-world" settings? In this extended abstract, we argue that the current de facto way of evaluating SSL

---

[*]Equal contribution
[†]Work done as a member of the Google Brain Residency Program (`https://g.co/brainresidency`)

| Method | CIFAR-10 4k Labels | SVHN 1k Labels |
|---|---|---|
| Π-M (Sajjadi et al., 2016b) | 11.29% | – |
| Π-M (Laine & Aila, 2017) | 12.36% | 4.82% |
| MT (Tarvainen & Valpola, 2017) | 12.31% | 3.95% |
| VAT (Miyato et al., 2017) | 11.36% | 5.42% |
| VAT + EM (Miyato et al., 2017) | 10.55% | 3.86% |
| **Results above this line cannot be directly compared to those below** | | |
| Supervised | $20.26 \pm 0.38\%$ | $12.83 \pm 0.47\%$ |
| Π-Model | $16.37 \pm 0.63\%$ | $7.19 \pm 0.27\%$ |
| Mean Teacher | $15.87 \pm 0.28\%$ | $5.65 \pm 0.47\%$ |
| VAT | $13.86 \pm 0.27\%$ | $5.63 \pm 0.20\%$ |
| VAT + EM | $13.13 \pm 0.39\%$ | $5.35 \pm 0.19\%$ |
| Pseudo-Label | $17.78 \pm 0.57\%$ | $7.62 \pm 0.29\%$ |

Table 1: Test error rates obtained by various SSL approaches on the standard benchmarks of CIFAR-10 with all but 4,000 labels removed and SVHN with all but 1,000 labels removed. Top: Reported results in the literature; Bottom: Using our proposed unified reimplementation. "Supervised" refers to using only 4,000 and 1,000 labeled datapoints from CIFAR-10 and SVHN respectively without any unlabeled data. Π-M, MT, VAT, PL, and EM refer to Π-Model, Mean Teacher, Virtual Adversarial Training, Pseudo-Labeling, and Entropy Minimization respectively (see appendix A). Note that the model below the line has roughly half as many parameters as most models above.

techniques does not address this question in a satisfying way. Specifically, the standard evaluation procedure of taking a large labeled dataset and discarding many of the labels fails to consider some common characteristics of SSL applications. We address this question by proposing a new experimental methodology which we believe better measures applicability to real-world problems.

## 2 IMPROVED EVALUATION

In this work, we make several improvements to the conventional experimental procedures used to evaluate SSL methods, which proceeds as follows: First, take a common (typically image classification) dataset used for supervised learning and throw away the labels for most of the dataset. Then, treat the portion of the dataset whose labels were retained as a small labeled dataset $\mathcal{D}$ and the remainder as an auxiliary unlabeled dataset $\mathcal{D}_{UL}$. Some (not necessarily standard) model is then trained and accuracy is reported using the unmodified test set. The choice of dataset and number of retained labels is somewhat standardized across different papers. Below, we enumerate some ways that we believe this procedure insufficiently reflects real-world applicability. We focus on each concern by carrying a series of experiments.

**A Shared Implementation.** We introduce a shared implementation of the underlying architectures used to compare all of the SSL methods that we discuss in appendix A, as described in appendix B.1. This improves over prior work because though the datasets used across different studies have largely become standardized over time, experimental details vary significantly. For every SSL technique, in addition to a "fully-supervised" (not utilizing unlabeled data) baseline, we ran 1000 trials of Gaussian Process-based black box optimization using Google Cloud Machine Learning's hyperparameter tuning service (Golovin et al., 2017). We optimized over hyperparameters specific to each SSL algorithm, in addition to those shared across approaches. We report the test error at the point of lowest validation error for the hyperparameters we chose, along with previously reported figures for these tasks, in table 1.

**High-Quality Fully-Supervised Baselines.** Do SSL methods work because of the unlabeled data itself? We examine this issue by designing a model with a regularization, data augmentation, and training scheme that gets close to the performance of SSL techniques without using any unlabeled data, as described in appendix B.3 Of course, comparing the performance of this model to SSL approaches applied to different models is unfair; however, we are interested in investigating the upper-bound of fully-supervised performance as a benchmark for future work. On 4000 labeled images from CIFAR-10, our best model obtained a test error of **13.4%**, averaged over 5 runs. **Since this model's performance is higher than that of some recent SSL techniques (Rasmus et al., 2015; Sajjadi et al., 2016a; Salimans et al., 2016), we argue that SSL algorithms must show that the use of unlabeled data produces a bigger performance boost than simply using better regularization and data augmentation.**

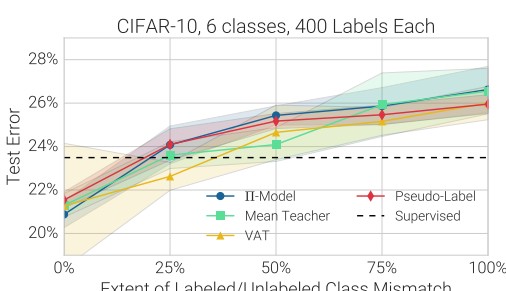

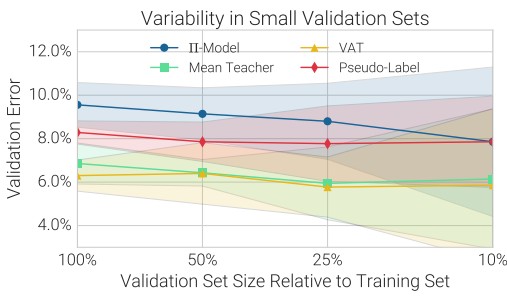

Figure 1: Test error for each SSL technique on CIFAR-10 (six animal classes) with a varying amount of overlap between classes in the labeled and unlabeled data. For example, "25%" refers to one of the four classes in the unlabeled data coming from a different class than the six in the labeled data. "Supervised" refers to using no unlabeled data.

Figure 2: Validation error over 10 randomly-sampled nonoverlapping validation sets of varying size. For each SSL approach, we re-evaluated an identical model on each randomly-sampled validation set. The mean and standard deviation of the validation error over the 10 validation sets are shown as lines and shaded regions respectively. Models are trained on SVHN with 1,000 labels. Validation set sizes are listed relative to the training size (e.g. 10% indicates a size-100 validation set).

**Transfer Learning.** In practice a common way to deal with limited data is to "transfer" a model trained on a separate, but similar, large labeled dataset (Donahue et al., 2014; Yosinski et al., 2014; Dai & Le, 2015). This is typically achieved by initializing the parameters of a new model with those from the original model, and "fine-tuning" this new model using the small dataset. While this approach is only feasible when an applicable source dataset is available, it nevertheless provides a powerful, widely-used, and rarely reported baseline to compare against. We apply this technique by pre-training our model on ImageNet (downsampled to 32x32) then fine-tuning on 4000 labeled images from CIFAR-10, as described in appendix B.4. The resulting model obtained an error rate of **12.0%** on the test set. **This is a lower error rate than any SSL technique achieved using this network, indicating that transfer learning may be a preferable alternative when a labeled dataset suitable for transfer is available.**

**Class Distribution Mismatch.** Note that when taking an existing fully-labeled dataset and discarding labels, all members of $\mathcal{D}_{UL}$ come from the same classes as those in $\mathcal{D}$. In contrast, consider the following example: Say you are trying to train a model to distinguish between one of ten faces, but you only have a few images for each of these ten faces. As a result, you augment your dataset with a large unlabeled dataset of images of random people's faces. In this case, it is extremely unlikely that any of the images in $\mathcal{D}_{UL}$ will be one of the ten people the model is trained to classify. We examine this issue by training with images from CIFAR-10, while controlling the level of overlap between the classes from which the labeled and unlabeled data were sampled, as described in appendix B.5. Our results are shown in fig. 1. **We demonstrate the surprising result that adding unlabeled data from a mismatched set of classes can actually *hurt* performance compared to not using any unlabeled data at all (points above the black dotted line in fig. 1).** However, we did not re-tune hyperparameters for each of these experiments; it is possible that adjusting hyperparameters in each setting could narrow this gap.

**Varying the Amount of Labeled *and* Unlabeled Data.** A somewhat common practice in SSL evaluation is to vary the number of labeled examples. We consider this, and the possibility of varying the number of unlabeled examples, in appendix B.6. **We find surisingly different levels of sensitivity to varying data amounts across SSL techniques.**

**Small Validation Sets.** In all of the experiments above (and in all experiments in the literature that we are aware of), hyperparameters are tuned on a labeled validation set which is significantly larger than the labeled portion of the training set. We are interested in measuring the extent to which this provides SSL algorithms with an unrealistic advantage, compared to real-world scenarios where the validation set would be smaller. A theoretical analysis and empirical process for considering this issue can be found in appendix B.7. **We find that for a realistically-sized validation set (10%**

**of the training set size), differentiating between the performance of the models is not feasible.**
This suggests that SSL methods which rely on heavy hyperparameter tuning on a large validation set
may have limited real-world success.

## 3 CONCLUSIONS AND RECOMMENDATIONS

Our experiments provide strong evidence that standard evaluation practice for SSL is unrealistic.
Based on our experimental analysis, in appendix C we recommend concrete changes to SSL evaluation
to better reflect real-world applications. SSL has seen a great streak of successes recently. We hope
that our results and unified implementation (code forthcoming) help push these successes towards the
real world.

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

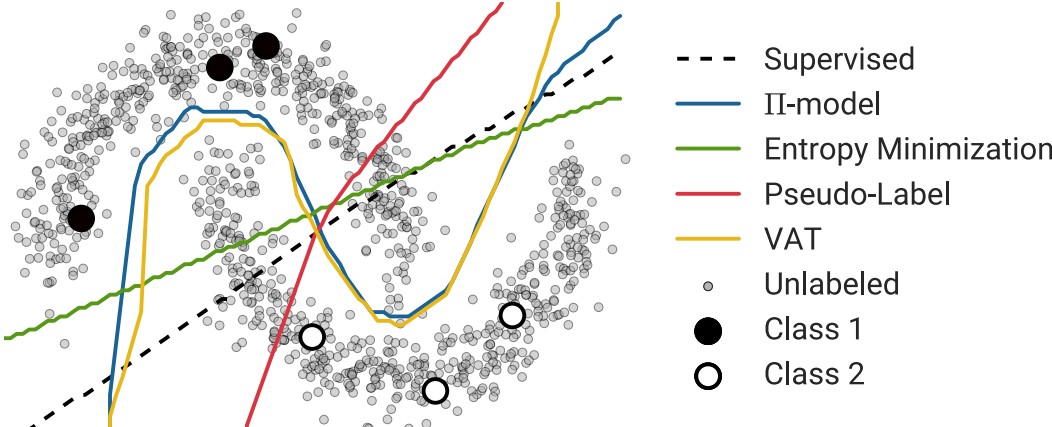

Figure 3: Demonstration of the behavior of different SSL approaches on a simple toy dataset ("two moons"). Each approach was applied to a neural network with three hidden layers, each with 10 units, all with ReLU nonlinearities. Training the network on only the labeled data (large black and white dots) produces a decision boundary (dashed line) which does not follow the contours of the data "manifold", as indicated by additional unlabeled data (small grey dots). In a simplified view, the goal of SSL is to leverage the unlabeled data to produce a decision boundary which better reflects the data.

## A    SEMI-SUPERVISED LEARNING METHODS

In supervised learning, we are given a training dataset of input-target pairs $(x, y) \in \mathcal{D}$ sampled from an unknown joint distribution $p(x, y)$. Our goal is to produce a prediction function $f_\theta(x)$ parametrized by $\theta$ which produces the correct target $y$ for previously unseen samples from $p(x)$. For example, choosing $\theta$ might amount to optimizing a loss function which reflects the extent to which $f_\theta(x) = y$ for $(x, y) \in \mathcal{D}$. In SSL we are additionally given a collection of unlabeled input datapoints $x \in \mathcal{D}_{UL}$. Now, we hope to also leverage the data from $\mathcal{D}_{UL}$ to help produce a prediction function which is more accurate than what would have been obtained by using $\mathcal{D}$ on its own.

From a broad perspective, the goal of SSL is to use $\mathcal{D}_{UL}$ to augment $f_\theta(x)$ with information about the structure of $p(x)$. For example, $\mathcal{D}_{UL}$ can provide hints about the shape of the data "manifold" which can produce a better estimate of the decision boundary between different possible target values. A depiction of this concept on a simple toy problem is shown in fig. 3, where the scarcity of labeled data makes the decision boundary between two classes ambiguous but the additional unlabeled data reveals clear structure which can be discovered by an effective SSL algorithm.

There have been a wide variety of methods proposed to achieve this goal, including "transductive" (Gammerman et al., 1998) variants of $k$-nearest neighbors (Joachims, 2003) and support vector machines (Joachims, 1999), graph-based methods (Zhu et al., 2003; Bengio et al., 2006), and algorithms based on learning features (frequently via generative modeling) from unlabeled data (Belkin & Niyogi, 2002; Lasserre et al., 2006; Salakhutdinov & Hinton, 2007; Coates & Ng, 2011; **?**; Kingma et al., 2014; Pu et al., 2016; Odena, 2016; Salimans et al., 2016). A comprehensive overview is out of the scope of this paper; we instead refer interested readers to (Zhu et al., 2003; Chapelle et al., 2006). Instead, we focus on the class of methods which solely involve adding an additional loss term to the training of an existing model, and otherwise leave the training and model unchanged from what would be used in the fully-supervised setting. We limit our focus to these approaches for the pragmatic reasons that they are simple to describe and implement and that they are currently the state-of-the-art for SSL on image classification datasets.

### A.1    CONSISTENCY REGULARIZATION

Consistency regularization describes a class of methods with following intuitive goal: Realistic perturbations $x \to \hat{x}$ of data points $x \in \mathcal{D}_{UL}$ should not change the output of $f_\theta(x)$. Generally, this involves minimizing $d(f_\theta(x), f_\theta(\hat{x}))$ where $d(\cdot, \cdot)$ measures a distance between the prediction function's outputs, e.g. mean-squared error or Kullback-Leibler divergence. Typically the gradient

through this consistency term is only backpropagated through $f_\theta(\hat{x})$. In the toy example of fig. 3, this would ideally result in the classifier effectively separating the two class clusters due to the fact that their members are all close together. This simple principle has produced a series of approaches which are currently state-of-the-art for SSL.

### A.1.1 STOCHASTIC PERTURBATIONS/Π-MODEL

The simplest setting in which to apply consistency regularization is when the prediction function $f_\theta(x)$ is itself stochastic, i.e. it can produce different outputs for the same input $x$. This is quite common in practice during training when $f_\theta(x)$ is a neural network due to common regularization techniques such as data augmentation, dropout, and adding noise. These regularization techniques themselves are typically designed in such a way that they ideally should not cause the model's prediction to change, and so are a natural fit for consistency regularization.

The straightforward application of consistency regularization is thus minimizing $d(f_\theta(x), f_\theta(\hat{x}))$ for $x \in \mathcal{D}_{UL}$ where in this case $d(\cdot, \cdot)$ is chosen to be mean squared error. This distance term is added to the classification loss as a regularizer, scaled by a weighting hyperparameter. This idea was simultaneously proposed in (Sajjadi et al., 2016b) and (Laine & Aila, 2017), referred to as "Regularization With Stochastic Transformations and Perturbations" and the "Π-Model" respectively. We adopt the latter name for its conciseness. In fig. 3, the Π-Model successfully finds the correct decision boundary.

### A.1.2 TEMPORAL ENSEMBLING/MEAN TEACHER

A difficulty with the Π-model approach is that it relies on a potentially unstable "target" prediction, namely the second stochastic network prediction which can rapidly change over the course of training. As a result, (Tarvainen & Valpola, 2017) and (Laine & Aila, 2017) proposed two methods for obtaining a more stable target output $\bar{f}_\theta(x)$ for $x \in \mathcal{D}_{UL}$. "Temporal Ensembling" (Laine & Aila, 2017) uses an exponentially accumulated average of outputs of $f_\theta(x)$ for the consistency target. Inspired by this approach, "Mean Teacher" (Tarvainen & Valpola, 2017) instead proposes to use a prediction function parametrized by an exponentially accumulated average of $\theta$ over training. As with the Π-model, the distance term $d(f_\theta(x), \bar{f}_\theta(x))$ is added as a regularization term with a weighting hyperparameter. In practice, it was found that the Mean Teacher approach outperformed Temporal Ensembling (Tarvainen & Valpola, 2017), so we will focus on it in our later experiments.

### A.1.3 VIRTUAL ADVERSARIAL TRAINING

Instead of relying on the built-in stochasticity of $f_\theta(x)$, Virtual Adversarial Training (VAT) (Miyato et al., 2017) directly approximates a tiny perturbation $r_{adv}$ to add to $x$ which would most significantly affect the output of the prediction function. The perturbation can be computed efficiently as

$$r \sim \mathcal{N}\left(0, \frac{\xi}{\sqrt{\dim(x)}} I\right) \tag{1}$$

$$g = \nabla_r d(f_\theta(x), f_\theta(x + r)) \tag{2}$$

$$r_{adv} = \epsilon \frac{g}{||g||} \tag{3}$$

where $\xi$ and $\epsilon$ are scalar hyperparameters. Consistency regularization is then applied to minimize $d(f_\theta(x), f_\theta(x + r_{adv}))$ with respect to $\theta$, effectively using the "clean" output as a target given an adversarially perturbed input. VAT is inspired by adversarial examples (Szegedy et al., 2014; Goodfellow et al., 2015), which are natural datapoints $x$ which have a virtually imperceptible perturbation added to them which causes a trained model to misclassify the datapoint. Like the Π-Model, the perturbations caused by VAT find the correct decision boundary in fig. 3.

### A.2 ENTROPY-BASED

A simple loss term which can be applied to unlabeled data is to encourage the network to make "confident" (low-entropy) predictions for all examples, regardless of the actual class predicted. Assuming a categorical output space with $K$ possible classes (e.g. a $K$-dimensional $\mathrm{softmax}$ output),

this gives rise to the "entropy minimization" term (Grandvalet & Bengio, 2005):

$$-\sum_{k=1}^{K} f_\theta(x)_k \log f_\theta(x)_k \tag{4}$$

Ideally, entropy minimization will discourage the decision boundary from passing near data points where it would otherwise be forced to produce a low-confidence prediction. However, given a high-capacity model, another valid low-entropy solution is simply to create a decision boundary which has overfit to locally avoid a small number of data points, which is what appears to have happened in the synthetic example of fig. 3. On its own, entropy minimization has not been shown to produce competitive results compared to the other methods described here (Sajjadi et al., 2016a). However, entropy minimization was combined with VAT to obtain state-of-the-art results by (Miyato et al., 2017). An alternative approach which is applicable to multi-label classification was proposed by (Sajjadi et al., 2016a), but it performed similarly to entropy minimization on standard "one-hot" classification tasks. Interestingly, entropy *maximization* was also proposed as a regularization strategy for neural networks by (Pereyra et al., 2017).

### A.3 Pseudo-Labeling

Pseudo-labeling (Lee, 2013) is a simple heuristic which is widely used in practice, likely because of its simplicity and generality – all that it requires is that the model provides a probability value for each of the possible labels. It proceeds by producing "pseudo-labels" for $\mathcal{D}_{UL}$ using the prediction function itself over the course of training. Pseudo-labels which have a corresponding class probability which is larger than a predefined threshold are used as targets for a standard supervised loss function applied to $\mathcal{D}_{UL}$. While intuitive, it can nevertheless produce incorrect results when the prediction function produces unhelpful targets for $\mathcal{D}_{UL}$, as shown in fig. 3. Note that pseudo-labeling is quite similar to entropy regularization, in the sense that it encourages the model to produce higher-confidence (lower-entropy) predictions for data in $\mathcal{D}_{UL}$ (Lee, 2013). However, it differs in that it only enforces this for data points which already had a low-entropy prediction due to the confidence thresholding. Pseudo-labeling is also closely related to self-training (Rosenberg et al., 2005), which differs only in the heuristics used to decide which pseudo-labels to retain.

## B Full Experimental Details

### B.1 Reproduction

We create a unified reimplementation of the methods outlined in appendix A using a common model architecture and training procedure. Note that our goal is *not* to produce state-of-the-art results, but instead to provide a rigorous comparative analysis in a common framework. Further, because our model architecture and training hyperparameters differ from those used to test SSL methods in the past, our results are not directly comparable to past work and should therefore be considered in isolation. We use this reimplementation as a consistent testbed on which we carry out a series of experiments, each of which individually focusses on a single issue from section 2.

For our reimplementation, we selected a standard model that is modern, widely used, and would be a completely reasonable choice for a practitioner working on image classification. This ideally avoids the possibility of using an architecture which is custom-tailored to work well with one particular SSL technique. We chose a Wide ResNet (Zagoruyko & Komodakis, 2016), due to their widespread adoption and availability. Specifically, we used "WRN-28-2", i.e. ResNet with depth 28 and width 2, including the standard batch normalization (Ioffe & Szegedy, 2015) and leaky ReLU nonlinearities (Maas et al., 2013). We did not deviate from the standard specification for WRN-28-2 – our model was virtually identical to the standard implementation in the `tensorflow/models` repository[1] – so we refer to (Zagoruyko & Komodakis, 2016) for model specifics. For training, we chose the ubiquitous Adam optimizer (Kingma & Ba, 2014). For all datasets, we followed standard procedures for regularization, data augmentation, and preprocessing; details are in appendix B.2.

Given the model, we implemented each of the SSL approaches in appendix A. We attempted to closely follow existing codebases whenever authors made them available. To ensure that all of the

---

[1] https://github.com/tensorflow/models/

techniques we are studying are given fair and equal treatment, and that we are reporting the best-case performance under our model, we carried out a large-scale hyperparameter optimization. For every SSL technique, in addition to a "fully-supervised" (not utilizing unlabeled data) baseline, we ran 1000 trials of Gaussian Process-based black box optimization using Google Cloud Machine Learning's hyperparameter tuning service (Golovin et al., 2017). We optimized over hyperparameters specific to each SSL algorithm, in addition to those shared across approaches.

We tested each SSL approach on the widely-reported image classification benchmarks of SVHN (Netzer et al., 2011) with all but 1000 labels discarded and CIFAR-10 (Krizhevsky, 2009) with all but 4,000 labels discarded. We optimized hyperparameters to minimize classification error on the standard validation set from each dataset, as is standard practice (an approach we evaluate critically in appendix B.7). Black-box hyperparameter optimization can produce unintuitive hyperparameter settings which vary unnecessarily between different datasets and SSL techniques. We therefore audited the best solutions found for each dataset/SSL approach combination and hand-designed a simpler, unified set of hyperparameters whose performance did not drastically differ from the best performance found by Vizier. An enumeration of these hyperparameter settings can be found in appendix B.1.1.

We report the test error at the point of lowest validation error for the hyperparameter settings we chose, along with previously reported figures for these tasks, in table 1. Note that our numbers cannot be directly compared to those previously reported due to a lack of a shared underlying network architecture. For example, our model has roughly half as many parameters as the one used in (Laine & Aila, 2017; Miyato et al., 2017; Tarvainen & Valpola, 2017), which may partially explain its somewhat worse performance. However, our findings are generally consistent with what has been reported in the literature; namely, that all of these SSL methods improve (to a varying degree) over the baseline. Further, Virtual Adversarial Training and Mean Teacher both appear to work best, which is consistent with their shared state-of-the-art status. We will use this default hyperparameter setting in all of the experiments that follow.

### B.1.1 HYPERPARAMETERS

In our hyperparameter search, for each SSL method, we always separately optimized algorithm-agnostic hyperparameters such as the learning rate, its decay schedule and weight decay coefficients. In addition, we optimized to those hyperparameters specific to different SSL approaches separately for each approach. In keeping with our argument in appendix B.7, we attempted to find hyperparameter settings which were performant across datasets and SSL approaches so that we could avoid unrealistic tweaking. After hand-tuning, we used the hyperparameter settings summarized in table 2, which lists those settings which were shared and common to all SSL approaches.

We trained all networks for 500,000 updates with a batch size of 100. We did not use any form of early stopping, but instead continuously monitored validation set performance and report test error at the point of lowest validation error. All models were trained with a single worker on a single GPU (i.e. no asynchronous training).

### B.2 DATASET DETAILS

Overall, we followed standard data normalization and augmentation practice. For SVHN, we converted image data to floating point values in the range [-1, 1]. For data augmentation, we solely used random translation by up to 2 pixels. We used the standard train/validation split, with 65,932 images for training and 7,325 for validation.

For any model which was to be used to classify CIFAR-10 (e.g. including the base ImageNet model for the transfer learning experiment in appendix B.4), we applied global contrast normalization and ZCA-normalized the inputs using statistics calculated on the CIFAR-10 training set. ZCA normalization is a widely-used and surprisingly important preprocessing step for CIFAR-10. Data augmentation on CIFAR-10 included random horizontal flipping, random translation by up to 2 pixels, and Gaussian input noise with standard deviation 0.15. We used the standard train/validation split, with 45,000 images for training and 5,000 for validation.

| Shared | |
|---|---|
| L1 regularization coefficient | 0.001 |
| L2 regularization coefficient | 0.0001 |
| Learning decayed by a factor of | 0.1 |
| at training iteration | 400,000 |
| Consistency coefficient rampup* | 200,000 |
| **$\Pi$-Model** | |
| Initial learning rate | 0.0003 |
| Max consistency coefficient | 20 |
| **Mean Teacher** | |
| Initial learning rate | 0.0003 |
| Max consistency coefficient | 50 |
| Exponential moving average decay | 0.99 |
| **VAT** | |
| Initial learning rate | 0.003 |
| Max consistency coefficient | 0.6 |
| VAT $\epsilon$ | 6.0 |
| VAT $\xi$ | $10^{-6}$ |
| **VAT + EM** (as for VAT) | |
| Entropy penalty multiplier | 0.1 |
| **Pseudo-Label** | |
| Initial learning rate | 0.003 |
| Max consistency coefficient | 1.0 |
| Pseudo-label threshold | 0.5 |

Table 2: Hyperparameter settings used in our experiments. All hyperparameters were tuned via large-scale hyperparameter optimization and then distilled to sensible and unified defaults by hand. Adam's $\beta_1$, $\beta_2$, and $\epsilon$ parameters were left to the defaults suggested by (Kingma & Ba, 2014). *Following (Tarvainen & Valpola, 2017), we ramped up the consistency coefficient starting from 0 to its maximum value using a sigmoid schedule so that it achieved its maximum value at 200,000 iterations.

## B.3 FULLY-SUPERVISED BASELINES

After extensive experimentation, we chose the large Shake-Shake model of (Gastaldi, 2017) due to its powerful regularization capabilities. We used a standard data-augmentation scheme consisting of random horizontal flips and random crops after zero-padding by 4 pixels on each side (He et al., 2016), as well as cutout regularization with a patch length of 16 pixels (DeVries & Taylor, 2017). Training and regularization was otherwise as in (Gastaldi, 2017), except we used a learning rate of 0.025 and a weight decay of 0.0025. On 4000 labeled images from CIFAR-10, this model obtained a test error of **13.4%**, averaged over 5 runs. This result emphasizes the importance of the underlying model to the evaluation of SSL algorithms, and reinforces our point that **different algorithms must be evaluated using the same model to avoid conflating comparison.**

## B.4 TRANSFER LEARNING

We trained our standard WRN-28-2 model on ImageNet (Deng et al., 2009) downsampled to 32x32 (Chrabaszcz et al., 2017) (the native image size of CIFAR-10). We used the same training hyper-parameters as used for the supervised baselines reported in appendix B.1. Then, we fine-tuned the model using 4,000 labeled data points from CIFAR-10.

## B.5 CLASS DISTRIBUTION MISMATCH

Now we examine the case where labeled and unlabeled data come from the same underlying distribution (e.g. natural images), but the unlabeled data contains classes not present in the labeled data.

To address this possibility, we synthetically vary the class overlap in our common test setting of CIFAR-10. Specifically, we perform 6-class classification on CIFAR-10's animal classes (bird, cat, deer, dog, frog, horse). The unlabled data comes from four classes — we vary how many of those four are among the six labeled classes to modulate class distribution mismatch. For completeness we also trained a model using no unlabeled data (fully supervised) and another model with the entire dataset as unlabeled data. We continue in the custom of using 400 labels per class for CIFAR-10, resulting in 2400 labeled examples.

Our results are shown in fig. 1. **We demonstrate the surprising result that adding unlabeled data from a mismatched set of classes can actually *hurt* performance compared to not using any unlabeled data at all (points above the black dotted line in fig. 1).** This implies that it may be preferable to pay a much larger cost to obain labeled data than to obtain unlabeled data if the unlabeled data is sufficiently unrelated to the core learning task. However, we did not re-tune hyperparameters for each of these experiments; it is possible that adjusting hyperparameters in each setting could narrow this gap.

## B.6 Varying Data Amounts

Many SSL techniques are tested only in the core settings we have studied so far, namely CIFAR-10 with 4,000 labels and SVHN with 1,000 labels. However, we argue that varying the amount of labeled data tests the extent to which performance degrades in the very-limited-label regime, and also at which point the approach can recover the performance of training with all of the labels in the dataset. We therefore ran experiments on both SVHN and CIFAR with different labeled data amounts; the results are shown in fig. 4. Notably, the performance of all of the SSL techniques tends to converge as the number of labels grows. Pseudo-labeling and the $\Pi$-Model seem to struggle in the very-few-label regime, whereas Mean Teacher performed well.

Another possibility is to vary the amount of unlabeled data. However, using the CIFAR-10 and SVHN datasets in isolation places an upper limit on the amount of unlabeled data available. Fortunately, SVHN is distributed with the "SVHN-extra" dataset, which adds 531,131 additional digit images and which was previously used as unlabeled data in (Tarvainen & Valpola, 2017). Similarly, the "Tiny Images" dataset can augment CIFAR-10 with eighty million additional unlabeled images as done in (Laine & Aila, 2017), however it also introduces a class distribution mismatch between labeled and unlabeled data because its images are not necessarily from the classes covered by CIFAR-10. As a result, we do not consider Tiny Images for auxiliary unlabeled data in this paper.

We evaluated the performance of each SSL technique on SVHN with 1,000 labels and varying amounts of unlabeled data from SVHN-extra, which resulted in the test errors shown in fig. 5. As expected, increasing the amount of unlabeled data improves the performance of SSL techniques. More surprising was the extent to which these gains differed between different approaches. For example, the amount of unlabeled data had an outsized impact on Mean Teacher, which for 10,000 unlabeled datapoints had the worst performance but vastly outperformed every other method when larger amounts of unlabeled data was available. This finding, coupled with Mean Teacher's robustness to small amounts of labeled data, suggests Mean Teacher may be a particularly good choice when labels are expensive but unlabeled data is plentiful. More broadly, **we find surisingly different levels of sensitivity to varying data amounts across SSL techniques.**

## B.7 Small Validation Sets

In all of the experiments above (and in all experiments in the literature that we are aware of), hyperparameters are tuned on a labeled validation set which is significantly larger than the labeled portion of the training set. We are interested in measuring the extent to which this provides SSL algorithms with an unrealistic advantage, compared to real-world scenarios where the validation set would be smaller. We can derive a theoretical estimate for the number of validation samples required to confidently differentiate between the performance of different approaches using Hoeffding's inequality (Hoeffding, 1963):

$$\mathbf{P}(|\bar{V} - \mathbb{E}[V]| \geq p) \leq 2\exp(-2np^2) \tag{5}$$

where in our case $\bar{V}$ is the empirical estimate of the validation error, $\mathbb{E}[V]$ is its hypothetical true value, $p$ is the desired maximum deviation between our estimate and the true value, and $n$ is the

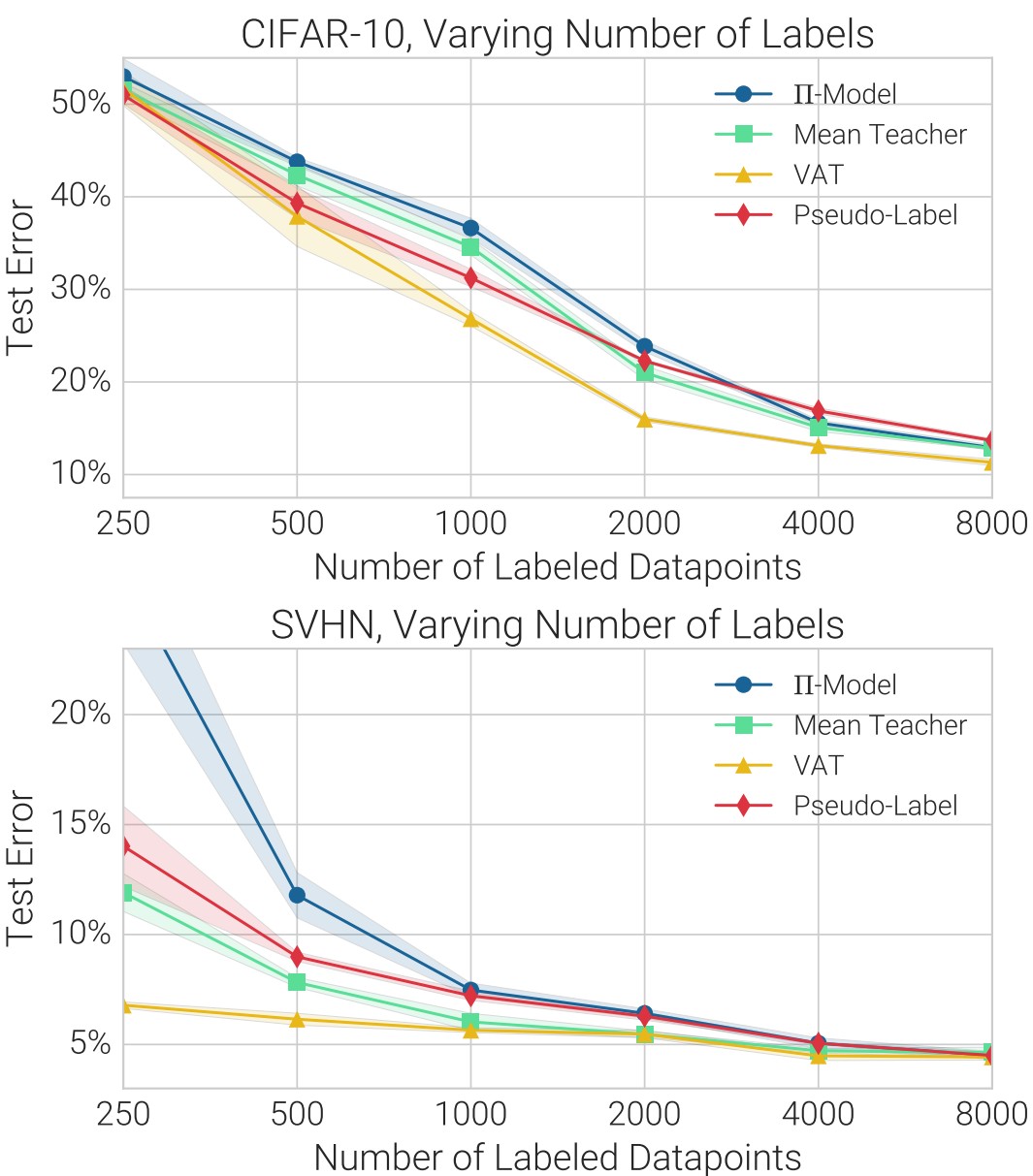

Figure 4: Test error for each SSL technique on SVHN and CIFAR-10 as the amount of labeled data varies.

number of examples in the validation set. In this analysis, we are treating validation error as the average of independent binary indicator variables denoting whether a given example in the validation set is classified correctly or not. As an example, if we want to be 95% confident that our estimate of the validation error differs by less than 1% absolute of the true value, we would need nearly 20,000 validation examples. This is a disheartening estimate due to the fact that the difference in test error achieved by different SSL algorithms reported in table 1 is often close to or smaller than 1%, but 20,000 is many times more samples than are provided in the training sets.

This theoretical analysis may be unrealistic due to the assumption that the validation accuracy is the average of independent variables. To measure this phenomenon empirically, we took baseline models trained with each SSL approach on SVHN with 1,000 labels and evaluated them on validation sets with varying sizes. These synthetic small validation sets were sampled randomly and without overlap

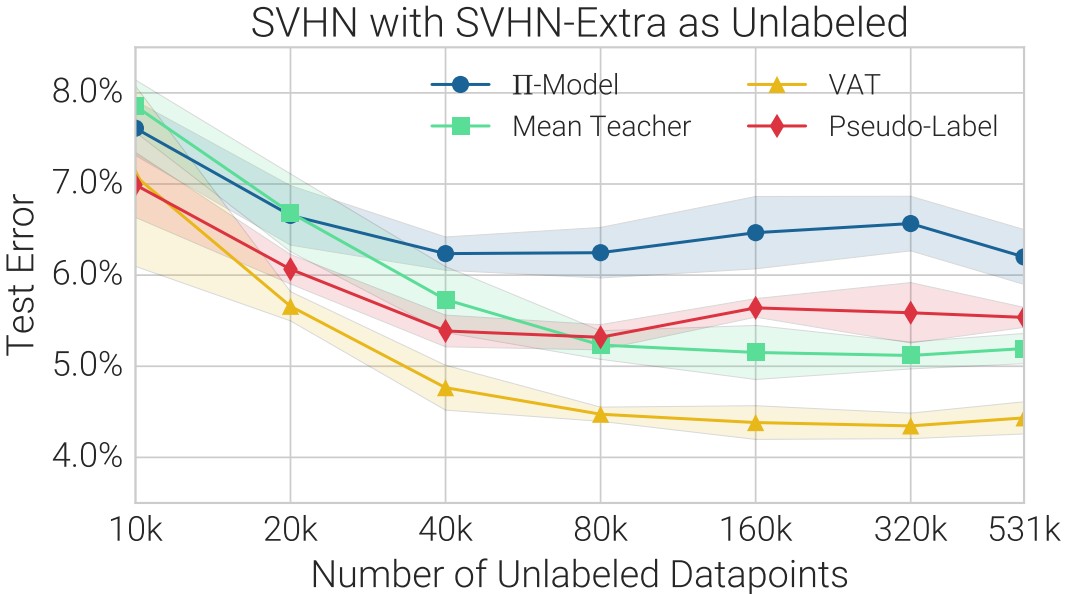

Figure 5: Test error for each SSL technique on SVHN with 1,000 labels and varying amounts of unlabeled images from SVHN-extra.

from the full SVHN validation set. We show the mean and standard deviation of validation error over 10 randomly-sampled validation sets for each of the models in fig. 2.

## C    RECOMMENDATIONS

What changes to SSL evaluation should be made to better reflect real-world applications? Our recommendations are as follows:

• Use the exact same underlying model when comparing SSL approaches. Differences in model structure or even implementation details can greatly impact results.

• Report carefully-tuned fully-supervised accuracy and transfer learning performance where applicable as baselines. The goal of SSL should be to significantly outperform the fully-supervised settings. A truly impressive method could outperform transfer learning as well.

• Report results where the class distribution mismatch systematically varies. We showed that the SSL techniques we studied all suffered when the unlabeled data came from different classes than the labeled data — a realistic scenario that to our knowledge is drastically understudied.

• Vary both the amount of labeled and unlabeled data when reporting performance. An ideal SSL algorithm is effective even with very little labeled data and benefits from additional unlabeled data. Specifically, we recommend combining SVHN with SVHN-Extra to test performance in the large-unlabeled-data regime.

• Take care not to over-tweak hyperparameters on an unrealistically large validation set. A SSL method which has hyperparameters that must be significantly changed on a per-model or per-task basis in order to achieve satisfactory performance will not be useable when validation sets are realistically small.

