# OpenReview forum: "Realistic Evaluation of Semi-Supervised Learning Algorithms"
_ICLR.cc/2018/Workshop — Accept_

### Official Review · AnonReviewer1 · 2018-03-05
**Good paper. Proposes framework for comparative analysis of Semi-supervised learning algorithms.**

**Rating:** 7
**Confidence:** 4

**Review:**

This paper proposes a framework and a shared implementation for comparison of various SSL-based approaches. The paper focuses on the class of SSL methods that augment the objective function with a regularizer based on unlabeled data e.g. manifold regularization, entropy regularization etc.

The main contributions of the paper are:

1). A novel framework (and implementation) for making apples-to-apples comparisons between different SSL algorithms.
2). Teasing out the real value of unlabeled data by comparison against fully-supervised baselines. The paper further dives deep into the impact of the class distribution of unlabeled data being different from labeled dataset, as well as the impact of the size of the validation data.

The paper is well written and does a nice job of reviewing the latest SSL work (especially in vision) and their experimental shortcomings. The SSL literature should pay heed to these in future work.

I think this paper can be further improved if the authors were to propose *quantifiable* improvement guidelines for the SSL community. This will help the SSL community build on this paper.

For example: 1). How much more labeled data (say as a % of the training  dataset) would this problem need so that a class of fully-supervised algorithms gives the same accuracy as a class of SSL algorithms?

2). What's the maximum class-distribution mismatch in terms of (say) the KL divergence, that adding unlabeled data would still give a performance boost?

These would provide concrete numbers which can be quoted in future SSL research.

---

### Official Review · AnonReviewer3 · 2018-03-08
**A good paper**

**Rating:** 7
**Confidence:** 4

**Review:**

This paper points out several current problems in evaluating semi-supervised learning (SSL) methods.
1. With a suitable regularization, data augmentation and training scheme, a model can approach state-of-the-art performance without using any unlabeled samples;
2. The class distribution of unlabeled samples is extremely important for models, which is not emphasized in previous works;
3. There is no unified evaluation of method behaviours in different amount of labeled and unlabeled data, which is also important for models in realistic settings;
4. Hyperparameters are tuned in unrealistically large validation sets.

Some principles and standards are provided for improving future research in this field, including,
1. unifying the underlying model;
2. comparing with strong baselines, like carefully-tuned fully-supervised models;
3. analyzing method's behaviour with different class distribution and amount of unlabeled samples.

In general, the paper may have significant impact in future evaluations of SSL methods for image classification tasks. At least it raises valuable questions about the evaluation of SSL methods, which is worth discussing.

My only concern is its writing quality and organization, which can be further improved. Potential problems and corresponding improvements are listed as follows,

* Appendix B.1.1: Almost all experiment are conducted based on a set of hand-tuned hyperparameters, which is claimed to be performant across datasets and SSL approaches (Appendix B.1.1). Since this set of hyperparameters can become a standard setting in future works, it is better to give evidence that it is a good choice, e.g. report the performance gap between this set of hyperparameters and the best hyperparameter settings among search results.

* Section 2, Class Distribution Mismatch: It is claimed that 'adjusting hyperparameters in each setting could narrow this gap', which is not confirmed by experiments. Although it is not very probable that the hurt of performance is caused by the wrong set of hyperparameters, it is possible.

* It is better to provide open-source version of the reimplementation in order to benefit future research in this field.

* Appendix B.3: It may not be a good style to state points in appendix.

* Since the paper only evaluates SSL algorithms for image classification tasks, it is better to change the title to make it less general, e.g.  'Realistic Evaluation of Semi-Supervised Learning Algorithms for Image Classification Task'.

* Minor typos: 'surisingly' -> 'surprisingly' in section 2 'Varying the Amount of Labeled and Unlabeled Data'.

---

### Decision · Program_Chairs · 2018-03-20
**ICLR 2018 Workshop Acceptance Decision**

**Decision:**

Accept

**Comment:**

Congratulations, your paper was accepted to the ICLR workshop.